# Learning-to-Cache:
# Accelerating Diffusion Transformer via Layer Caching

**Xinyin Ma**[1]   **Gongfan Fang**[1]   **Michael Bi Mi**[2]   **Xinchao Wang**[1*]

National University of Singapore[1]   Huawei Technologies Ltd.[2]

maxinyin@u.nus.edu, xinchao@nus.edu.sg

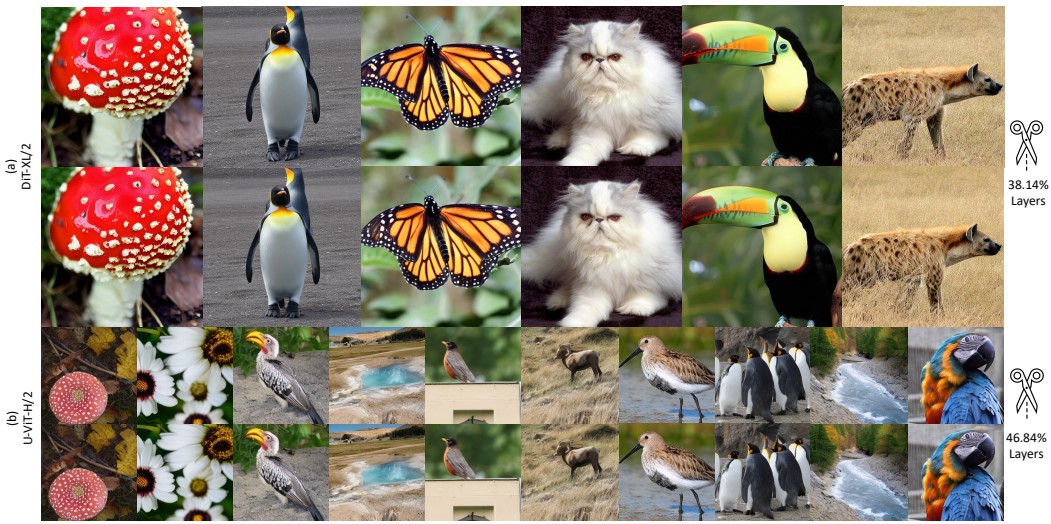

Figure 1: (a) Generate 512×512 images using DiT-XL/2, sampled by DDIM with 50 NFEs. (b) Generate 256×256 images using U-ViT-H/2, sampled by DPM-Solver-2 with 50 NFEs.

## Abstract

Diffusion Transformers have recently demonstrated unprecedented generative capabilities for various tasks. The encouraging results, however, come with the cost of slow inference, since each denoising step requires inference on a transformer model with a large scale of parameters. In this study, we make an interesting and somehow surprising observation: the computation of a large proportion of layers in the diffusion transformer, through introducing a caching mechanism, can be readily removed even without updating the model parameters. In the case of U-ViT-H/2, for example, we may remove up to 93.68% of the computation in the cache steps (46.84% for all steps), with less than 0.01 drop in FID. To achieve this, we introduce a novel scheme, named **L**earning-to-**C**ache (L2C), that learns to conduct caching in a dynamic manner for diffusion transformers. Specifically, by leveraging the identical structure of layers in transformers and the sequential nature of diffusion, we explore redundant computations between timesteps by treating each layer as the fundamental unit for caching. To address the challenge of the exponential search space in deep models for identifying layers to cache and remove, we propose a novel differentiable optimization objective. An input-invariant yet timestep-variant router is then optimized, which can finally produce a static computation graph. Experimental results show that L2C largely outperforms samplers such as DDIM and DPM-Solver, alongside prior cache-based methods at the same inference speed.

*Corresponding author

38th Conference on Neural Information Processing Systems (NeurIPS 2024).

# 1 Introduction

In recent years, diffusion models [60, 59, 19] have achieved remarkable performance as powerful generative models for image generation [49, 10]. Among the various backbone designs for diffusion models, transformers [62] have emerged as a strong contender, showing its exceptional capabilities not only in synthesizing high-fidelity images [46, 3] but also in video generation [38, 7, 4], text-to-speech synthesis [33, 21, 61] and 3D generation [41, 5]. The diffusion transformer, while benefiting greatly from the great property of scalability of the transformer architecture, however, also brings about significant challenges in efficiency, including high deployment costs and slow inference speed.

Since the cost of sampling increases proportionally with the number of timesteps and the model size per timestep, naturally, current methods for increasing the sampling efficiency entail two branches: reducing the sampling steps[57, 19, 34, 2] or reducing the inference cost per step [16, 67]. The methods to reduce the number of timesteps include distilling the trajectory into fewer steps [52, 58, 39], discretizing the reverse-time SDE or the probability flow ODE [57, 73, 37]. Methods in another branch are mainly about compressing the model size [25, 30] or using a low-precision data format [18, 53]. A new method in the dynamic inference of diffusion is a special cache mechanism in the denoising process [40, 64]. These methods leverage the high similarity between the two steps and the special property of U-Net to cache some of the computations, which would be directly used in the next step. Some other dynamic inference methods employ a spectrum of diffusion models and allocate different networks for different steps [65, 44].

Previous approaches, especially those aimed at reducing model size, have predominantly targeted the compression of the U-Net architecture [50]. Our objective is to explore a paradigm for inference acceleration that is more suitable for transformer-based diffusion models. Unlike other architectures, transformers are distinctively composed of several layers with consistent structure. Based on this property, previous compression work on transformers mainly focuses on layer pruning [71] and random layer dropping [14, 48], as optimizing at the layer level tends to achieve higher speedup ratios compared to width pruning [24, 15, 8]. However, for diffusion transformers, we observed that dropping layers without retraining is not feasible. Removing even a few layers significantly degrades image quality (see Section 4.3). This observation highlights that the redundancy among layers at varying depths is not evident in DiT. Therefore, we consider another perspective of redundancy: the redundancy across layers situated at the same depths but occurring at different timesteps.

Motivated by cache-based methods [40, 64, 28], we aim to explore the existence and limitations of layer redundancy between timesteps within the diffusion transformer. A straightforward approach involves an exhaustive search where each layer is either cached or not, resulting in an exponentially growing search space with the depth of the layers. Additionally, heuristic-based layer selection cannot adequately address the mutual dependencies between layers. To overcome these challenges, we designed a framework that makes the problem of layer selection differentiable. Specifically, we interpolate predictions between two adjacent steps. This interpolation spans two extremes: a fast configuration where all layers are cached at the expense of image quality, and a slow configuration where all layers are retained, achieving optimal performance. We then search this interpolation space to identify an optimal caching scheme, optimizing a specialized router. This router is time-dependent but input-invariant, allowing the creation of a static computation graph for inference. We train this router by formulating an optimization problem that does not require updating model parameters, making it both cost-effective and easy to optimize.

Our results indicate that different percentages of layers can be cached in DiT [41] and U-ViT [3]. Notably, for U-ViT-H/2 on ImageNet, approximately 93.68% of layers are cacheable in the cache step, whereas for DiT-XL/2, the cacheable ratio is 47.43%, both with an almost negligible performance loss ($\Delta$FID < 0.01). By comparison, with the same acceleration ratio, a sampler with fewer steps would compromise image quality. Our method L2C can significantly outperform the fast sampler, as well as previous cache-based methods. Additionally, we observed distinct sparsity patterns for layers between these two models, suggesting significant behavioral variations between different architecture designs for diffusion transformers.

In summary, our contribution is the proposal of a novel acceleration method, learning-to-cache (L2C), specifically for diffusion transformers. We convert the non-differentiable layer selection problem into a differentiable optimization problem by interpolation, facilitating the learning of layer caching. Our results demonstrate that a large proportion of layers in the diffusion trans-

former can be cached without compromising performance. Furthermore, our approach significantly outperforms samplers with fewer steps and other cache-based methods. The code is available at https://github.com/horseee/learning-to-cache

## 2 Related Work

**Transformers in Diffusion Models.** Diffusion models have demonstrated broad applicability across various domains[13, 4, 69]. Transformer [62] is applied in diffusion models as an alternative to UNet[50]. GenViT[68] integrates the ViT[12] architecture into DDPM. U-ViT [3] employs the long skip connections between shallow and deep layers. DiT [46] shows the scalability of diffusion transformers and is further used as a general architecture for text-to-video generation [4, 38], speech synthesis [33] and 3D generation [5].

**Acceleration of Diffusion Models.** Generating images by diffusion models requires several rounds of model evaluation which is time-expensive. Some works focus on reducing the number of sampling steps in a training-free manner. DDIM[57] extends the original DDPM to non-Markovian cases. DPM-Solver[36, 37] further approximates the solution of diffusion ODES by the exponential integrators. EDM[23] finds that the Heun'2 2nd order method provides an excellent tradeoff between truncation error and NFE. More works try to solve either SDEs[60, 22, 11] or ODEs[34, 73, 72] in a more accurate and fast way. Other training-based methods [52, 31] distill and half the sampling steps. [58, 39] learns to map any point on the ODE trajectory to its origin. Another line of work reduces the workload per step. The model per step is compressed by reducing the parameter size [16, 6, 71, 63], using reduced precision [29, 18] and re-design the structure of the diffusion model [67, 30, 75, 25, 35]. In addition to static model inference, dynamic model inference has also been extensively explored within diffusion models, which employs different models for inference at varying steps. [32, 45] switch between different sizes of models in a model zoo. [42] designs a time-dependent exit schedule to skip a subset of parameters. Other works focus on denoising diffusion models in parallel[9], either through iterative optimization[54] or image splitting[27]. In addition to inference acceleration, some works also show how to train a diffusion model more efficiently by employing different training paradigm [17, 76] or from the data perspective [47].

**Cache in Diffusion Models** Cache [55] is used in computer systems to hold temporarily those portions of contents in the main memory which is believed to be used in a short time. Recently, [40, 64, 1] explores the cache mechanism in diffusion models. Based on the observations that the similarities of high-level features [70] is typically very high in consecutive steps, they propose to reuse the feature maps. By utilizing the computation flow of U-Net, [40] reuse the high-level features while updating the low-level features. [64, 28] further discovers the better position in U-Net to be cached. [20] proposes to reuse the attention map. [64, 56, 40] adjust the lifetime for each caching features and [64] further scales and shifts the reused features. [74] finds the cross-attention is redundant in the fidelity-improving stage and can be cached. [66] hashes and caches the images rendered from camera positions and diffusion timesteps to improve the efficiency of 3D generative modeling.

## 3 Method

### 3.1 Preliminary

The forward diffusion process starts at the starting point $\boldsymbol{x}_0$, where $\boldsymbol{x}_0$ is sampled from the data distribution $q(\boldsymbol{x}_0)$ to be learned. $\boldsymbol{x}_0$ is degenerated with gradually added Gaussian noise, with:

$$\boldsymbol{x}_t \sim q(\boldsymbol{x}_t|\boldsymbol{x}_0) = \mathcal{N}\left(\boldsymbol{x}_t; \alpha_t \boldsymbol{x}_0, \sigma_t^2 \mathbf{I}\right) \tag{1}$$

where $\alpha_t$ and $\sigma_t$ is the noise coefficient. We can quickly sample $x_t$ at arbitrary timestep by reparameterization trick. And for the reverse process, given two timesteps $s$ and $t$, where $s > 0$ and $t < s$, $x_t$ is calculated as[36]:

$$\boldsymbol{x}_t = \frac{\alpha_t}{\alpha_s} \boldsymbol{x}_s - \alpha_t \int_{\lambda_s}^{\lambda_t} e^{-\lambda} \hat{\boldsymbol{\epsilon}}_\theta \left(\boldsymbol{x}_{t_\lambda(\lambda)}, t_\lambda(\lambda)\right) \mathrm{d}\lambda \tag{2}$$

where $\lambda_t = \log\left(\alpha_t/\sigma_t\right)$. $t_\lambda(\lambda)$ is the inverse function of $\lambda_t$ that satisfies $t_\lambda(\lambda_t) = t$. $\boldsymbol{\epsilon}_\theta\left(\cdot\right)$ often represents the learned model, which, in our case, is the diffusion transformer. Previous methods

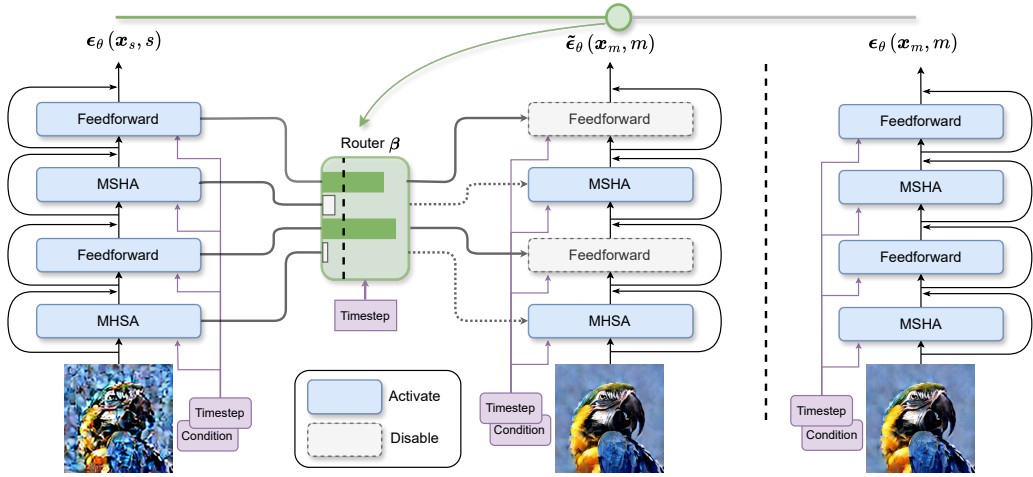

Figure 2: Illustration of Learning-to-Cache. When a layer is activated, the calculation proceeds as usual. In contrast, when a layer is disabled, the computation of the non-residual path is bypassed, and the results from the previous step are utilized instead. The router $\boldsymbol{\beta}$ smoothly controls the transition between two endpoints $\boldsymbol{\epsilon}_\theta(\boldsymbol{x}_s, s)$ and $\boldsymbol{\epsilon}_\theta(\boldsymbol{x}_m, m)$.

show that this integral term can be approximated by adopting Taylor expansion at $\lambda_s$, adopting the first-order [57] or higher-order approximation of this [36]. Take the first-order one as an example, the update of $\boldsymbol{x}_t$ would be:

$$\boldsymbol{x}_t = \frac{\alpha_t}{\alpha_s}\boldsymbol{x}_s - \sigma_t \left(e^{\lambda_t - \lambda_s} - 1\right)\boldsymbol{\epsilon}_\theta\left(\boldsymbol{x}_s, s\right) \tag{3}$$

### 3.2 Approximating $\boldsymbol{\epsilon}_\theta(\cdot)$ with a lightweight substitute

The question falls into how to efficiently calculate the term $\int_{\lambda_s}^{\lambda_t} e^{-\lambda}\hat{\boldsymbol{\epsilon}}_\theta\left(\boldsymbol{x}_{t_\lambda(\lambda)}, t_\lambda(\lambda)\right) \mathrm{d}\lambda$. Our core idea is that we want to keep more updates between $s$ and $t$ while the overall inference time would not increase too much. Suppose that we have three timesteps: $s$ and $t$ and one step $m$ between $s$ and $t$, the calculation of $\boldsymbol{x}_t$, in the case of Eq.3, would become:

$$\boldsymbol{x}_t = \frac{\alpha_t}{\alpha_m}\boldsymbol{x}_m - \sigma_t \left(e^{\lambda_t - \lambda_m} - 1\right)\boldsymbol{\epsilon}_\theta\left(\boldsymbol{x}_m, m\right), \text{ where } \boldsymbol{x}_m = \frac{\alpha_m}{\alpha_s}\boldsymbol{x}_s - \sigma_m \left(e^{\lambda_m - \lambda_s} - 1\right)\boldsymbol{\epsilon}_\theta\left(\boldsymbol{x}_s, s\right) \tag{4}$$

If we directly set $\boldsymbol{\epsilon}_\theta\left(\boldsymbol{x}_m, m\right) = \boldsymbol{\epsilon}_\theta\left(\boldsymbol{x}_s, s\right)$, it would be equivalent to the results in Equation 3 if we take a step directly from $s$ to $t$ (see the derivation in Appendix A.1). This approach results in faster computation, as it eliminates the need to compute $\boldsymbol{\epsilon}_\theta\left(\boldsymbol{x}_m, m\right)$; however, it compromises the quality of the resulting image. In contrast, another time-consuming but optimal way is to calculate $\boldsymbol{\epsilon}_\theta\left(\boldsymbol{x}_m, m\right)$ as usual, which necessitates a full model evaluation but yields superior image quality.

Recognizing that $\boldsymbol{\epsilon}_\theta\left(\boldsymbol{x}_s, s\right)$ represents a rapid yet suboptimal solution and $\boldsymbol{\epsilon}_\theta\left(\boldsymbol{x}_m, m\right)$ represents a slower but optimal solution when calculating $\boldsymbol{x}_t$, we want to find a model $\tilde{\boldsymbol{\epsilon}}(\boldsymbol{x}_m, m)$, which is the interpolation of these two models. We first define the interpolation as follows:

$$\tilde{\boldsymbol{\epsilon}}_\theta\left(\boldsymbol{x}_m, m; \boldsymbol{\beta}\right) = \mathcal{I}(\boldsymbol{\epsilon}_\theta\left(\boldsymbol{x}_s, s\right), \boldsymbol{\epsilon}_\theta\left(\boldsymbol{x}_m, m\right), \boldsymbol{\beta}) \tag{5}$$

where $\tilde{\boldsymbol{\epsilon}}_\theta\left(\boldsymbol{x}_m, m\right)$ is controlled by a set of variables $\boldsymbol{\beta}$, functioning as a slider that can smoothly transition between the two endpoints $\boldsymbol{\epsilon}_\theta\left(\boldsymbol{x}_s, s\right)$ and $\boldsymbol{\epsilon}_\theta\left(\boldsymbol{x}_m, m\right)$. $\tilde{\boldsymbol{\epsilon}}_\theta\left(\boldsymbol{x}_m, m\right)$ needs to meet two criteria: it should approximate the output of $\boldsymbol{\epsilon}_\theta\left(\boldsymbol{x}_m, m\right)$ and be faster for inference compared to $\boldsymbol{\epsilon}_\theta\left(\boldsymbol{x}_m, m\right)$. By creating the interpolation $\mathcal{I}$, we generate a large collection of models, allowing us to search within this set to find if there exists an $\tilde{\boldsymbol{\epsilon}}_\theta(\boldsymbol{x}_m, m)$ that satisfies our requirements.

### 3.3 Caching the Layer: A Feasible Choice for the Interpolation $\mathcal{I}$

In this section, we specifically define an interpolation $\mathcal{I}$ and explore the possibility of the existence of $\tilde{\boldsymbol{\epsilon}}_\theta\left(\boldsymbol{x}_m, m\right)$ within it. Given the transformer architecture, we propose an interpolation schema

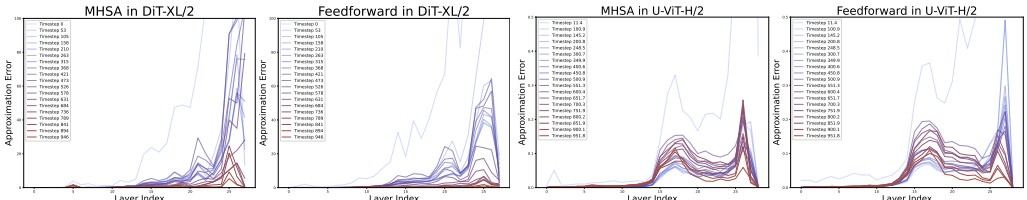

Figure 3: Approximation Error for DiT and U-ViT in different timesteps and different layers

by leveraging the layers of the transformer model. Here we take the computation of DiT[46] as an illustrative example. The transformer model can be decomposed into a sequence of basic layers $L_i(h,t)_{i=1}^D$, where $L_i(h,t) = h + g(t) * f_i(h,t)$, consisting of a residual connection. Here, $h$ is the input feature, and $D$ denotes the depth of the model. $t$ is the time condition. $f_i(h,t)$ can represent either a multi-head self-attention (MHSA) block or a pointwise feedforward block, and $g(t)$ is a time-conditioned scalar. We omit the condition $t$ in $f_i(h,t)$ for simplicity. Then we can construct a linear interpolation within the layers, and this interpolation of layer satisfies the model interpolation $\mathcal{I}$ (See Appendix A.2):

$$\tilde{L}_i(h_i^m, m; \alpha_i, \beta_i) = h_i^m - (1 - \alpha_i) \cdot (h_i^m - h_i^s) + g(m)(\beta_i \cdot f(h_i^m) + (1 - \beta_i) \cdot f(h_i^s)) \quad (6)$$

where $h_i^s$ and $h_i^m$ is the input to the block $L_i$ at timestep $s$ and $m$ respectively. $\beta_i$ is a coefficient in layer $i$ to control the proximity to $f(h_i^m)$ or $f(x_i^s)$ and $\alpha_i$ is to used as an control for the input. Both of these variables are constrained within the range $[0, 1]$.

This interpolation provides a special mechanism for inference. If $\beta_i$ in layer $i$ is set to 0, the output can be directly taken from the layer in the previous timestep, allowing the computation cost in this layer to be skipped. Non-zero $\beta_i$ would trigger the original computation of layer $i$. A discretized $\beta_i$ can be seen as a router, which selects the layers to be activated or disabled. And for $\alpha_i$, it can be set to any value since there is almost no computation cost for a combination of $h_i^m$ and $h_i^s$ and we choose $\alpha_i = 0$. By setting more $\beta_i$ in different layers to 0, the acceleration ratio can be cumulative. Therefore, we can calculate the total computational cost based on the number of non-zero $\beta_i$, and our goal $\tilde{\epsilon}_{\theta}(\boldsymbol{x}_m, m)$ can be interpreted as finding as many zeros in $\{\beta_i\}_{i=1}^D$ as possible with the minimal approximation error between $\tilde{\epsilon}_{\theta}(\boldsymbol{x}_m, m)$ and $\epsilon_{\theta}(\boldsymbol{x}_m, m)$.

**One key observation.** One greedy way for finding the $\beta_i$ in each layer is taking the approximation error of each layer into account:

$$E = ||\tilde{L}(\cdot) - L(\cdot)||_2^2 = (1 - \beta_i) \cdot |g(m)| \cdot ||f(h_i^m) - f(h_i^s)||_2^2 \quad (7)$$

and taking $\beta_i$ in those layer with smallest $|g(m)| \cdot ||f(h_i^m) - f(h_i^s)||_2^2$ to be 0. In Figure 3, we analyze $||f(h_i^m) - f(h_i^s)||_2^2$ in two types of models: DiT and U-ViT. We find that performance varies significantly across different timesteps, even at the same layer. Particularly in the DiT model, the error is markedly higher in the later steps compared to the early denoising steps. Additionally, the performance of multi-head self-attention differs substantially from that of feedforward layers. Based on this, we assign each timestep with its own $\{\beta_i\}_{i=1}^D$. Thus, $\boldsymbol{\beta}$ becomes time-variant, where $\boldsymbol{\beta} = \{\beta_{ij} \mid i = 1, 2, \ldots, T; j = 1, 2, \ldots, D\}$ and $T$ is the total denoising steps.

In addition, we directly use this metric as the criterion for $\beta_{ij}$ and employ it during inference. From the experimental results in 4, we observe that it cannot effectively handle a combination of layers. This limitation arises because the approximation error for each layer is influenced by changes in the preceding layer. However, exhaustively evaluating all possible configurations is impractical, as the number of trials increases exponentially with the depth of the model.

### 3.4 Learning to Cache

To address this, we propose the following method: Learning to Cache. Recall that our goal is to find a $\tilde{\epsilon}_{\theta}(\boldsymbol{x}_m, m)$ that is (1) as close as $\epsilon_{\theta}(\boldsymbol{x}_m, m)$ and (2) with minimal computation cost. We can reformulate this as an optimization problem as:

$$\arg\min_{\boldsymbol{\beta}} ||\tilde{\epsilon}(\boldsymbol{x}_m, m; \boldsymbol{\beta}) - \epsilon(\boldsymbol{x}_m, m)||_2^2 \quad \text{s.t.} \quad \sum_{i=1}^D \delta_{\beta_{ij}1} \leq C \quad (8)$$

**Algorithm 1** Training

1: **Input:** Data distribution $p(\boldsymbol{x}_0)$, diffusion model $\epsilon_\theta(\cdot)$, learning rate $\eta$, ODE solver $\Psi(\cdot)$, total steps $T$ and the step schedule $\{t_i\}_{i=1}^T$ in $\Psi(\cdot)$
2: $\boldsymbol{\beta} \sim \mathcal{N}(0,1)$
3: **repeat**
4:     $\boldsymbol{x}_0 \sim p(\boldsymbol{x}_0), n \sim \mathcal{U}[1, T//2]$
    // Step $s$ for calculating states for caching
5:     $s \leftarrow t_{n*2}$
6:     $\boldsymbol{x}_s \sim \mathcal{N}(\boldsymbol{x}_s; \alpha_s \boldsymbol{x}_0, \sigma_s^2 \mathbf{I})$
7:     $\epsilon_s \leftarrow \epsilon_\theta(\boldsymbol{x}_s, s)$ and cache $\{f(\cdot)\}_{i=1}^D$
    // Step $m$ for using cached states
8:     $m \leftarrow t_{n*2-1}$
9:     $\boldsymbol{x}_m \leftarrow \Psi(\epsilon_s, s, m)$
10:     $\beta_m \leftarrow \mathrm{Sigmoid}(\boldsymbol{\beta}_m)$
    // Optimize
11:     Calculate $\tilde{\epsilon}(\boldsymbol{x}_m, m; \beta_m)$ by Eq.6
12:     $\mathcal{L} \leftarrow ||\tilde{\epsilon}(x_m, m) - \epsilon_\theta(x_m, m)||_2^2 + \lambda \sum \beta_m$
13:     $\boldsymbol{\beta}_m \leftarrow \boldsymbol{\beta}_m - \eta \nabla_{\boldsymbol{\beta}_m} \mathcal{L}$
14: **until** converged

**Algorithm 2** Sampling

1: **Input:** Diffusion model $\epsilon_\theta(\cdot)$, router $\boldsymbol{\beta}$, ODE solver $\Psi(\cdot)$, threshold $\theta$, total steps $T$ and the step schedule $\{t_i\}_{i=1}^T$ in $\Psi(\cdot)$
2: $\boldsymbol{x}_T \sim \mathcal{N}(\mathbf{0}, \mathbf{I})$
3: **for** $n = T, \ldots, 1$ **do**
4:     $h_1^{t_n} \leftarrow \boldsymbol{x}_n$
5:     **for** $i = 1, \ldots, D$ **do**
6:       **if** $\mathrm{Sigmoid}(\boldsymbol{\beta}_{t_n i}) > \theta$ and step $n$ is the cache step **then**
7:         $\beta_i \leftarrow 0$
8:       **else**
9:         $\beta_i \leftarrow 1$
10:       **end if**
11:       $h_{i+1}^{t_n} \leftarrow \tilde{L}_i(h_i^{t_n}, t_n; 0, \beta_i)$ by Eq.6
12:     **end for**
13:     $\tilde{\epsilon}(x_n, t_n) \leftarrow h_{D+1}^{t_n}$
14:     $\boldsymbol{x}_{n-1} \leftarrow \Psi(\tilde{\epsilon}(\boldsymbol{x}_n, t_n), t_n, t_{n-1})$
15: **end for**
16: **return** $\boldsymbol{x}_0$

where $C$ is the constraint for the total cost. $\delta_{\beta_{ij}1}$ is the Kronecker delta function, which is 1 if $\beta_{ij} = 1$. Though $\beta_{ij}$ in the final solution needs to be discrete, $\beta_{ij}$ is designed to be continuous to make the computation differentiable when optimized. And when inference, a threshold $\theta$ would be set to discretize the $\beta_{ij}$ to be either 0 or 1, where $\beta_{ij}$ turned to become a router. The only trained variables in our algorithm are $\boldsymbol{\beta}$. Thus, the parameters in the diffusion model would remain unchanged. With the help of Lagrange duality to transform the optimization problem into an unconstrained one, the loss would be:

$$\mathcal{L}(\tilde{\boldsymbol{\epsilon}}, \boldsymbol{\epsilon}, \boldsymbol{x}_m, m; \boldsymbol{\beta}) = ||\tilde{\epsilon}(\boldsymbol{x}_m, m; \boldsymbol{\beta}) - \epsilon(\boldsymbol{x}_m, m)||_2^2 + \lambda \cdot \sum_{i=1}^D \beta_{ij} \tag{9}$$

where $\lambda$ is the Lagrangian multiplier that governs the regularization. We show the algorithm for training and inference in Algorithm 1 and 2. To ensure $\boldsymbol{\beta}$ remains within the range $[0, 1]$, a sigmoid operation is performed before $\boldsymbol{\beta}$ is passed into the model. In these algorithms, we adopt layer caching every two steps, representing that only half of the steps would inference in a faster speed. For simplicity, the image encoder and decoder are omitted.

## 4 Experiments

### 4.1 Experimental Setup

**Models and Datasets.** We explore our methods on two commonly used transformer architectures in diffusion models: DiT [46] and U-ViT [3]. Specifically, we use DiT-XL/2 (256×256), DiT-XL/2 (512×512), DiT-L/2 and U-ViT-H/2. Except for DiT-L/2, we use the officially released models. We trained a DiT-L/2 for one million steps, which is used to investigate if layer redundancy exists in smaller models that may not be fully converged. Most of the results are presented under the resolution 256×256 and we also show the results on models that generate high resolution 512×512 images.

**Implementations.** Since the parameters of the diffusion model would not be updated, the only parameters that require optimization are $\boldsymbol{\beta}$, resulting in a very limited number of variables. For example, for DiT-XL-2 with 20 denoising steps, the number of trainable variables is 560. We take the training set of ImageNet to train $\beta$ for 1 epoch. The learning rate is set to 0.01 and AdamW optimizer is used to optimize $\beta$. The training is conducted upon 8 A5000 GPUs with a global batch size equal to 64. To train with classifier-free guidance, we randomly drop some labels and assign a null token to the label. The dropping rates for labels follow the original training pipeline.

Table 1: Accelerating image generation on ImageNet for the DiT model family.

| Methods | NFE | MACs(T) | Latency(s) | Speedup | IS↑ | FID↓ | sFID↓ | Precision↑ | Recall↑ |
|---|---|---|---|---|---|---|---|---|---|
| | | | | DiT-XL/2 (ImageNet 256×256) (cfg=1.5) | | | | | |
| DDPM | 250 | 28.61 | 36.55 | - | 280.1 | 2.27 | 4.54 | 82.73 | 57.95 |
| DDIM | 250 | 28.61 | 36.45 | - | 243.4 | 2.14 | 4.55 | 80.70 | 60.57 |
| DDIM | 50 | 5.72 | 7.25 | 1.00× | 238.6 | 2.26 | 4.29 | 80.16 | 59.89 |
| DDIM | 40 | 4.57 | 5.82 | 1.24× | 239.8 | 2.39 | 4.28 | 80.36 | **59.13** |
| Ours | 50 | 4.36 | 5.57 | 1.30× | **244.1** | **2.27** | **4.23** | **80.94** | 58.76 |
| DDIM | 20 | 2.29 | 2.87 | 1.00× | 223.5 | 3.48 | 4.89 | 78.76 | 57.07 |
| DDIM | 16 | 1.83 | 2.30 | 1.25× | 210.9 | 4.68 | 5.71 | 76.78 | **56.20** |
| Ours | 20 | 1.78 | 2.26 | 1.27× | **227.0** | **3.46** | **4.64** | **79.15** | 55.62 |
| DDIM | 10 | 1.14 | 1.43 | 1.00× | 158.3 | 12.38 | 11.22 | 66.78 | 52.82 |
| DDIM | 9 | 1.03 | 1.29 | 1.11× | 140.9 | 16.57 | 14.21 | 62.28 | 49.98 |
| Ours | 10 | 1.04 | 1.30 | 1.10× | **156.3** | **12.79** | **10.42** | **66.21** | **52.15** |
| | | | | DiT-XL/2 (ImageNet 512×512) (cfg=1.5) | | | | | |
| DDIM | 50 | 22.85 | 37.73 | 1.00× | 204.1 | 3.28 | 4.50 | 83.33 | 54.80 |
| DDIM | 30 | 13.71 | 22.51 | 1.68× | 198.3 | 3.85 | **4.92** | **83.01** | **56.00** |
| Ours | 50 | 14.19 | 22.57 | 1.67× | **202.1** | **3.69** | 5.03 | 82.90 | 54.60 |
| | | | | DiT-L/2 (ImageNet 256×256) (cfg=1.5) | | | | | |
| DDIM | 50 | 3.88 | 5.06 | 1.00× | 167.6 | 4.82 | 4.40 | 78.72 | 54.66 |
| DDIM | 40 | 3.10 | 4.06 | 1.25× | 168.2 | 4.99 | 4.43 | **79.01** | 54.71 |
| Ours | 50 | 2.95 | 4.01 | 1.26× | **168.3** | **4.82** | **4.41** | 78.97 | **54.73** |
| DDIM | 20 | 1.55 | 2.01 | 1.00× | 160.16 | 6.45 | 5.26 | 77.13 | 53.65 |
| DDIM | 16 | 1.24 | 1.63 | 1.23× | 151.70 | 7.91 | 6.24 | 75.93 | 51.71 |
| Ours | 20 | 1.20 | 1.60 | 1.26× | **160.53** | **6.55** | **5.08** | **77.47** | **52.22** |

Table 2: Results with U-ViT-H/2 on ImageNet dataset. The resolution here is 256×256. We adopt the DPM-Solver-2, which has 2 function evaluations per step. The total NFE (instead of steps) is reported below. Guidance strength is set to 0.4.

| Methods | NFE | MACs | Latency | Speedup | FID↓ | NFE | MACs | Latency | Speedup | FID↓ |
|---|---|---|---|---|---|---|---|---|---|---|
| DPM-Solver | 50 | 6.44 | 19.37 | 1.00× | 2.3728 | 20 | 2.58 | 7.69 | 1.00× | 2.5739 |
| DPM-Solver | 30 | 3.86 | 11.55 | 1.68× | 2.4644 | 16 | 2.06 | 6.08 | 1.26× | 2.7005 |
| Ours | 50 | 3.79 | 11.16 | 1.74× | **2.3625** | 20 | 1.92 | 5.64 | 1.35× | **2.5809** |

**Evaluation.** We tested our method upon two samplers, DDIM[57] and DPM-Solver[36], with sampling steps from 10 to 50. For the DiT model, we use the DDIM sampler. And for U-ViT, we use the DPM-Solver-2. All the experiments here use classifier-free guidance. To evaluate the image quality, 50k images are generated per trial. We measure the image quality with Frechet Inception Distance(FID)[43], sFID[43], Inception Score[51], Precision and Recall[26]. Besides, we reported the total MACs and the latency to make a comparison of the acceleration ratio. The MACs is evaluated using pytorch-OpCounter[2], and the latency is tested when generating a batch of images(8 images) with classifier-free guidance on a single A5000, which we conducted five tests and took the average.

## 4.2 Main Results

We present the results of DiT in Tables 1 and 2, comparing our algorithms with samplers of comparable inference speed. Our method requires more denoising steps, but each step takes less average time. In contrast, samplers require fewer steps, but each step takes more time. Our experiments demonstrate that our methods significantly outperform DDIM and DPM-Solver. For instance, with the 20-step DDIM on DiT-XL/2, our method achieves an FID of 3.46, nearly identical to the unaccelerated one. In comparison, the DDIM achieves an FID of 4.68. When generating high-resolution images, sampling with fewer steps, or using a relatively smaller model, our method still outperforms baselines.

---

[2]https://github.com/Lyken17/pytorch-OpCounter

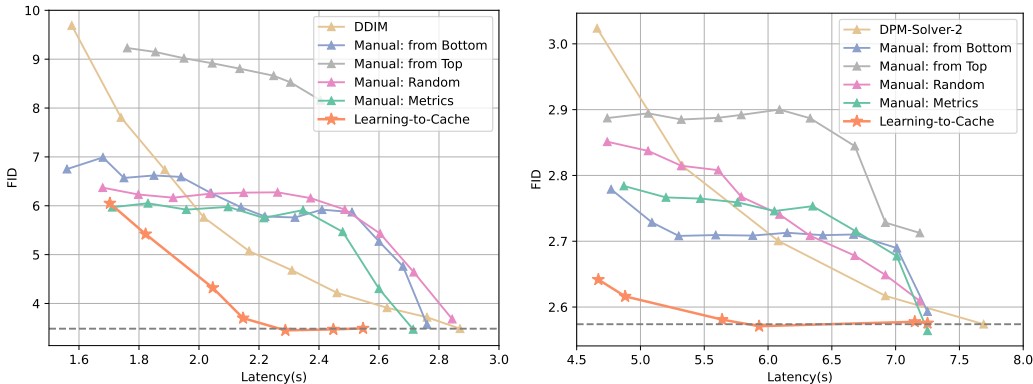

Figure 4: Speed-Quality Tradeoff for DiT-XL/2 and U-ViT-H/2 with 20 denosing steps as the basis. The dashed line indicates the performance without applying inference acceleration.

Table 3: Comparison with other cache-based method on U-ViT

| Methods | NFE | Latency | Speedup | FID↓ |
|---|---|---|---|---|
| DPM-Solver | 20 | 7.69 | 1.00× | 2.57 |
| DeepCache[40] | 20 | 4.68 | 1.64× | 2.70 |
| Ours | 20 | 4.62 | 1.67× | **2.64** |
| Faster Diffusion[28] | 20 | 5.95 | 1.29× | 2.82 |
| Ours | 20 | 5.93 | 1.30× | **2.57** |

Table 4: Maximum cacheable layers for DiT and U-ViT with different steps.

| Model | DiT-XL/2 | | U-ViT-H/2 | |
|---|---|---|---|---|
| NFE | 50 | 20 | 50 | 20 |
| Remove Ratio | 47.43% | 44.29% | 93.68% | 63.79% |
| FFN Remove Ratio | 47.85% | 44.64% | 94.11% | 60.54% |
| MHSA Remove Ratio | 47.00% | 43.93% | 93.25% | 67.05% |

However, we observe that achieving nearly lossless compression under these conditions is challenging. We argue that this difficulty arises because layer redundancy is less apparent in these scenarios.

**Quality-Latency Tradeoff.** We show the trade-off curve between FID and Latency in Figure 4. These figures offer a more comprehensive comparison with two types of baselines: (1) **Heuristic Methods for Selecting Layers**. We designed several methods for selecting layers to cache, including rule-based approaches such as caching from top to bottom or from bottom to top, randomly selecting layers, and metric-based selection as described in Eq.7. We found that when the dependency between layers must be considered, they fail to select the optimal layers, leading to a degradation in image quality. In contrast, our method consistently achieves improved quality across various acceleration ratios. (2) **Sampler with fewer steps**. Our method significantly outperforms DDIM and DPM-Solver, as evidenced by the detailed comparison provided.

**Maximum Cacheable Layers for diffusion transformer.** From the trade-off curve, we found that there exists an upper limit for the number of cacheable layers. Below this limit, image quality remains almost unaffected, as indicated by a FID degradation of less than 0.01. This limit is detailed in Table 4. Notably, caching does not occur at every step: step $s$ involves full model inference, while only step $m$ caches layers. With a significant proportion of layers can be cached and the computation of these layers to be saved, notable differences emerge between the U-ViT and DiT models. For instance, in U-ViT, up to 94% of layers can be discarded for the cache step during the denoising process, whereas this proportion is considerably lower for DiT. Furthermore, we observed that the cacheable ratios for FFN and MHSA vary.

**Comparison with other cache-based methods** We also compared our method with other cache-based methods. Notably, previous cache-based methods are strongly coupled to the U-Net structure and cannot be applied to models without the U-structure, such as DiT. To ensure a fair comparison, we selected U-ViT, which incorporates both the U-structure and transformers, to implement these methods as baselines alongside our method. Table 3 presents the comparison results. The findings demonstrate that our method achieves better quality than the baselines.

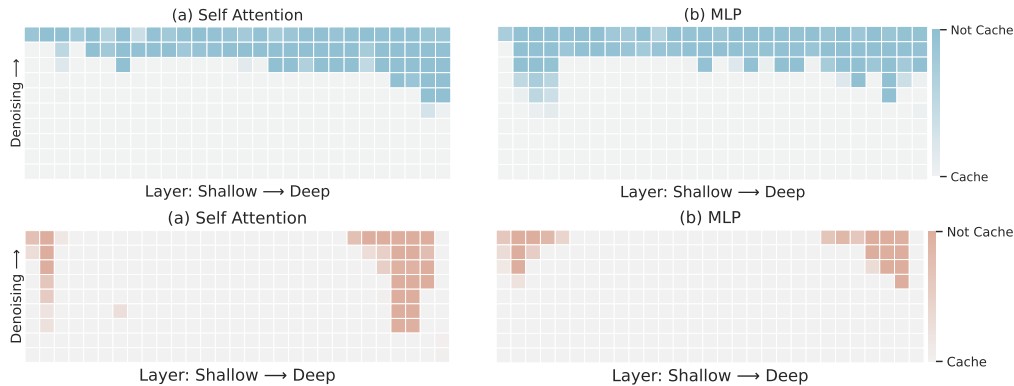

Figure 5: Learned Router $\beta$ for DiT-XL/2 (Top) and U-ViT-H/2 (Bottom). Different caching patterns are observed in different types of diffusion transformers.

Table 5: Comparison with layer dropout. The removal ratio corresponds to the percentage of sublayers being removed, including both MHSA and MLP blocks, for a total of 28 layers and 10 steps.

| Methods | Remove Ratio | Latency(s) | Speedup | IS↑ | FID↓ | sFID↓ | Precision↑ | Recall↑ |
|---|---|---|---|---|---|---|---|---|
| Random Drop | 170/560 | 2.439 | 1.18× | 3.36 | 277.42 | 171.83 | 1.23 | 0.24 |
| Learning-to-Drop | 179/560 | 2.421 | 1.19× | 113.93 | 17.35 | 28.46 | 60.25 | 52.68 |
| Learning-to-Cache | 176/560 | 2.438 | 1.18× | 226.13 | 3.47 | 4.58 | 79.19 | 56.47 |

## 4.3 Analysis

**The Learned Pattern of $\beta$**   We present the learned pattern in Figure 5. The two different architectures produce distinct patterns. For U-ViT, the entire middle section is almost entirely cacheable, allowing it to be replaced with the results from the previous step's calculations. However, the computations at both ends of the model are crucial and cannot be discarded. This observation explains why DeepCache outperforms faster-diffusion on U-ViT, as the learned patterns resemble the manually designed approach of DeepCache. However, this phenomenon is not clearly observed in DiT-XL. Additionally, we found a consistent tendency across models to retain more computation in the later stages while discarding calculations in the earlier stages. This observation aligns with our findings in Figure 3. When comparing the impact of different steps within the same layer, removing parts with smaller timestep has a greater effect on the changes in the output.

**Comparison between Layer Cache and Layer Dropout**   Layer dropout involves directly removing $f_i(\cdot)$, retaining only the computation in the skip path. We compare our method with layer dropout, where the layers are either randomly dropped or optimized using our algorithm (named Learning-to-Drop). The results, presented in Table 5, indicate that layer caching significantly outperforms layer dropout. Interestingly, when we learn the layers to be dropped, the models still produce acceptable images, although the quality is not as high. Illustrative examples are provided in Appendix B.2.

**Choice of threshold**   We investigated the effect of different thresholds on the image quality. Results are shown in Figure 6, where the model here is trained with six different $\lambda$ (corresponding to 6 points on one curve). We show the effect of different $\lambda$ in Appendix B.3. Our results reveal that for higher acceleration ratios, a larger threshold improves image quality. Conversely, for lower acceleration ratios, a smaller threshold is more effective. These also findings suggest that ranking layers by importance is not a reliable approach, since the selection of layers does not follow a strict sequential order. Otherwise, one threshold would win all.

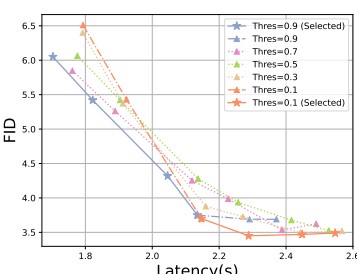

Figure 6: Effect of threshold $\theta$.

# 5  Limitation

The primary limitation of this work arises from its dependence on the trained diffusion models. For instance, when applied to DiT-XL/2 at a resolution of 512, our method encounters a slight drop in FID. Although it still surpasses the baseline, this indicates that the lossless caching of the layers does not uniformly exist across all models. It highlights significant variations between different models, and thus our method is strongly dependent on the structure design of the trained diffusion models. Another limitation of our method is that the acceleration is capped at $2\times$ because every two steps consist of one full model inference step and one cheaper step. This inherently restricts the maximum achievable acceleration ratio. However, we believe that this approach can be expanded to more than two steps, potentially improving the overall efficiency.

# 6  Conclusion

In this paper, we propose a novel acceleration method for diffusion transformers. By interpolating between the computationally inexpensive solution but suboptimal model, and the optimal solution but expensive model, we find there exist some models which would infer much faster and also produce high-fidelity images. To find this we train the router which is continuous when training and would be discretized when inference. Experiments show that our method largely outperforms baselines such as DDIM, DPM-Solver and other cache-based methods.

## Acknowledgement

This project is supported by the National Research Foundation, Singapore, under its Medium Sized Center for Advanced Robotics Technology Innovation.

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

# A Proof

## A.1 Two equivalent solutions to obtain $x_t$

To got the solution of $\boldsymbol{x}_t$, the following two approaches yield equivalent results:

1. Directly update $\boldsymbol{x}_t$ from $\boldsymbol{x}_s$. By the definition, the solution at time $t$ would be:

$$\boldsymbol{x}_t = \frac{\alpha_t}{\alpha_s} \boldsymbol{x}_s - \sigma_t \left( e^{\lambda_t - \lambda_s} - 1 \right) \boldsymbol{\epsilon}_\theta \left( \boldsymbol{x}_s, s \right) \tag{10}$$

2. First compute $\boldsymbol{x}_m$ from $\boldsymbol{x}_s$, and then compute $\boldsymbol{x}_t$ from $\boldsymbol{x}_m$ with $\boldsymbol{\epsilon}_\theta \left( \boldsymbol{x}_m, m \right) = \boldsymbol{\epsilon}_\theta \left( \boldsymbol{x}_s, s \right)$

*Proof.* First, we consider the solution of $\boldsymbol{x}_m$ from $\boldsymbol{x}_s$:

$$\boldsymbol{x}_m = \frac{\alpha_m}{\alpha_s} \boldsymbol{x}_s - \sigma_m \left( e^{\lambda_m - \lambda_s} - 1 \right) \boldsymbol{\epsilon}_\theta \left( \boldsymbol{x}_s, s \right) \tag{11}$$

And for the calculation of $x_t$ with $\boldsymbol{\epsilon}_\theta \left( \boldsymbol{x}_m, m \right) = \boldsymbol{\epsilon}_\theta \left( \boldsymbol{x}_s, s \right)$, we have

$$
\begin{aligned}
\boldsymbol{x}_t &= \frac{\alpha_t}{\alpha_m} \boldsymbol{x}_m - \sigma_t \left( e^{\lambda_t - \lambda_m} - 1 \right) \boldsymbol{\epsilon}_\theta \left( \boldsymbol{x}_m, m \right) \\
&= \frac{\alpha_t}{\alpha_m} \left( \frac{\alpha_m}{\alpha_s} \boldsymbol{x}_s - \sigma_m \left( e^{\lambda_m - \lambda_s} - 1 \right) \boldsymbol{\epsilon}_\theta \left( \boldsymbol{x}_s, s \right) \right) - \sigma_t \left( e^{\lambda_t - \lambda_m} - 1 \right) \boldsymbol{\epsilon}_\theta \left( \boldsymbol{x}_s, s \right) \\
&= \frac{\alpha_t}{\alpha_s} \boldsymbol{x}_s - \left( \frac{\alpha_t}{\alpha_m} \sigma_m \left( e^{\lambda_m - \lambda_s} - 1 \right) + \sigma_t \left( e^{\lambda_t - \lambda_m} - 1 \right) \right) \boldsymbol{\epsilon}_\theta \left( \boldsymbol{x}_s, s \right)
\end{aligned}
\tag{12}
$$

Note that $\lambda_t = \log \left( \alpha_t / \sigma_t \right)$. We obtain:

$$
\begin{aligned}
\boldsymbol{x}_t &= \frac{\alpha_t}{\alpha_s} \boldsymbol{x}_s - \left( \frac{\alpha_t}{\alpha_m} \sigma_m \left( \frac{\alpha_m}{\sigma_m} \frac{\sigma_s}{\alpha_s} - 1 \right) + \sigma_t \left( \frac{\alpha_t}{\sigma_t} \frac{\sigma_m}{\alpha_m} - 1 \right) \right) \boldsymbol{\epsilon}_\theta \left( \boldsymbol{x}_s, s \right) \\
&= \frac{\alpha_t}{\alpha_s} \boldsymbol{x}_s - \left( \alpha_t \frac{\sigma_s}{\alpha_s} - \sigma_t \right) \boldsymbol{\epsilon}_\theta \left( \boldsymbol{x}_s, s \right) = \frac{\alpha_t}{\alpha_s} \boldsymbol{x}_s - \sigma_t \left( e^{\lambda_t - \lambda_s} - 1 \right) \boldsymbol{\epsilon}_\theta \left( \boldsymbol{x}_s, s \right)
\end{aligned}
\tag{13}
$$

## A.2 Layer interpolation and Interpolation $\mathcal{I}$

We next show that the following interpolation of the layer would satisfy the interpolation $\mathcal{I}$ between $\boldsymbol{\epsilon}_\theta \left( \boldsymbol{x}_s, s \right)$ and $\boldsymbol{\epsilon}_\theta \left( \boldsymbol{x}_m, m \right)$ as we define:

$$\tilde{L}_i(h_i^m, m) = h_i^m - (1 - \alpha_i) \cdot (h_i^m - h_i^s) + g(m) \left( \beta_i \cdot f(h_i^m) + (1 - \beta_i) \cdot f(h_i^s) \right) \tag{14}$$

To prove this, we need to show these three things: (1) Interpolation condition, where the function passes through the given two models $\boldsymbol{\epsilon}_\theta \left( \boldsymbol{x}_s, s \right)$ and $\boldsymbol{\epsilon}_\theta \left( \boldsymbol{x}_m, m \right)$; (2) Continuity, where the interpolation function is continuous and (3) Differentiability, where the function is differentiable. Since $\beta_i$ and $\alpha_i$ are continuous and the model also satisfies these conditions, the only thing that needs to be proved is the first property.

*Proof.* We show Eq.14 satisfies the interpolation condition of $\mathcal{I}$

- With $\{\alpha_i\}_{i=1}^D$ and $\{\beta_i\}_{i=1}^D$ set to 0, the output of the transformer would be $\boldsymbol{\epsilon}_\theta \left( \boldsymbol{x}_s, s \right)$
  If for $i \in (1, D)$, $\alpha_i = 0$ and $\beta_i = 0$ then

$$\tilde{L}_i(h_i^m, m) = h_i^s + g(m) \cdot f(h_i^s) \tag{15}$$

  The output of the transformer after $D$ layer is given by:

$$\tilde{L}_D \left( \tilde{L}_{D-1} \left( \ldots \tilde{L}_1 \left( \boldsymbol{x}_s, s \right) \ldots \right) \right) = \boldsymbol{\epsilon}_\theta \left( \boldsymbol{x}_s, \boldsymbol{s} \right) \tag{16}$$

  Therefor, we get $\boldsymbol{\epsilon}_\theta \left( \boldsymbol{x}_s, s \right)$, one of the endpoint in the interpolation $\mathcal{I}$.

- With $\{\alpha_i\}_{i=1}^D$ and $\{\beta_i\}_{i=1}^D$ set to 1, the output would be $\boldsymbol{\epsilon}_\theta \left( \boldsymbol{x}_m, m \right)$. If for $i \in (1, D)$, $\alpha_i = 1$ and $\beta_i = 1$ then

$$\tilde{L}_i(h_i^m, m) = h_i^m + g(m) \cdot f(h_i^m) \tag{17}$$

  The same as above, we would get $\boldsymbol{\epsilon}_\theta \left( \boldsymbol{x}_m, m \right)$, the other endpoint in the interpolation $\mathcal{I}$.

# B  Additional Experiments

## B.1  Shifted cache step for DPM-Solver

Table 6: DPM-Solver with and without Shifted Cache Steps. Here we cache all the layers.

| Method | NFE | Latency | Speedup | IS | FID | sFID | Precision | Recall |
|---|---|---|---|---|---|---|---|---|
| DPM-Solver-2 | 20 | 7.69 | 1.00× | 263.76 | 2.57 | 5.01 | 82.77 | 55.71 |
| Cache | 20 | 4.25 | 1.81× | 222.64 | 5.30 | 7.87 | 76.17 | 54.59 |
| Cache - shifted | 20 | 4.54 | 1.70× | 254.48 | 2.80 | 4.70 | 81.14 | 55.48 |

One important trick used in our experiment with DPM-Solver involves shifting the cache step. Specifically, when employing DPM-Solver-2, the cache steps (step here is the model evaluation) are shifted from [2,4,6,8,10,...] to [3,5,7,9,11,...]. This adjustment is necessary because the DPM-Solver-2 requires the first-order derivative of the model $\epsilon_\theta(\cdot)$ at the current timestep, which is computed by subtracting the output at timestep $i$ from the output at timestep $i + 1$. If the cache steps were taken at timestep $i + 1$, it would result in an incorrect estimation of the derivative. By shifting the cache step, we ensure the accurate calculation of the derivative of $\epsilon_\theta(\cdot)$. This adjustment significantly impacts the results, as demonstrated in Table 6.

## B.2  Layer Dropout v.s. Layer Cache

Here we present further comparisons between layer dropout and layer caching. As illustrated in Figure 7, layer caching significantly outperforms layer dropout, maintaining pixel-wise consistency with the original pipeline. Conversely, when the layers to be dropped are selected by our algorithm, the model can still generate images with correct semantics. However, randomly dropping layers severely compromises the model's ability to produce acceptable images. Table 7 demonstrates that even a small proportion of layer dropout (around 10%) results in a substantial performance degradation.

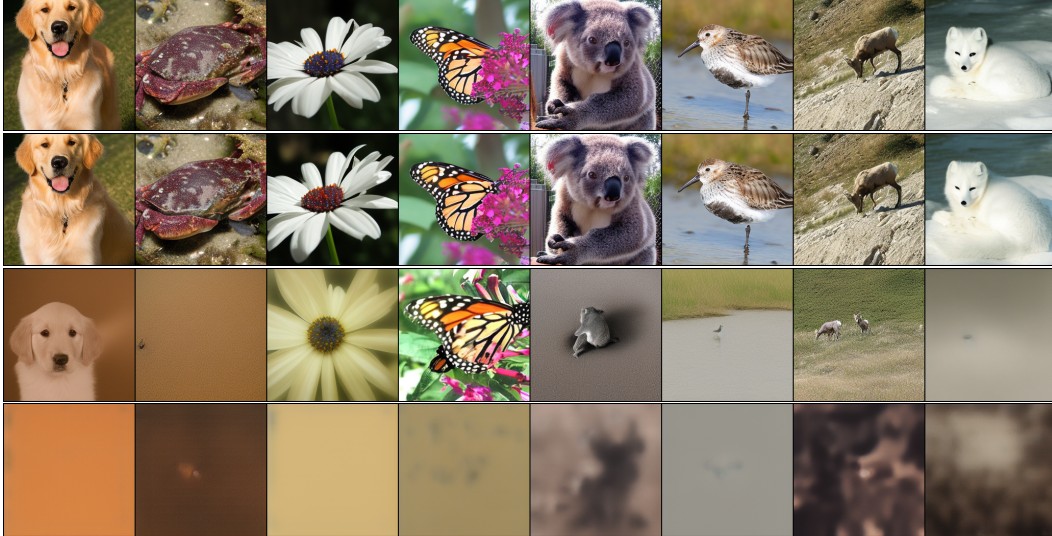

Figure 7: The quantitative results for layer dropping and layer caching in Section 4.3. (a) DDIM Pipeline with 20 NFE. (2) Our method L2C with 20 NFE (3) Learn to drop the layers by our algorithm. (4) Randomly drop layers. The results here, except the first line as the baseline, all speed up the inference by around 1.18×-1.19×.

## B.3  Effect of the hyper-parameter $\lambda$ and $\theta$

We find in our experiments that the router we learned is not sensitive to the hyper-parameters, including the learning rate, the training epoch, and the hyperparameters in the optimizer. The only

Table 7: Comparison with Layer Dropout

| Methods | Remove Ratio | Latency(s) | Speedup | IS↑ | FID↓ | sFID↓ | Precision↑ | Recall↑ |
|---|---|---|---|---|---|---|---|---|
| Random Drop | 60/560 | 2.718 | 1.06× | 9.66 | 112.93 | 153.48 | 10.56 | 65.57 |
| Random Drop | 170/560 | 2.439 | 1.18× | 3.36 | 277.42 | 171.83 | 1.23 | 0.24 |
| Learning-to-Drop | 179/560 | 2.421 | 1.19× | 113.93 | 17.35 | 28.46 | 60.25 | 52.68 |
| Learning-to-Cache | 176/560 | 2.438 | 1.18× | 226.13 | 3.47 | 4.58 | 79.19 | 56.47 |

Table 8: $\lambda$ and $\theta$ for training the router

| Model | DiT-XL/2 | DiT-XL/2 | DiT-XL/2 | DiT-XL/2 | DiT-L/2 | DiT-L/2 | U-ViT-H/2 | U-ViT-H/2 |
|---|---|---|---|---|---|---|---|---|
| NFE | 50 | 20 | 10 | 50 | 50 | 20 | 50 | 20 |
| Resolution | 256 | 256 | 256 | 512 | 256 | 256 | 256 | 256 |
| Sampler | DDIM | DDIM | DDIM | DDIM | DDIM | DDIM | DPM-Solver-2 | DPM-Solver-2 |
| $\lambda$ for train | 1e-6 | 5e-6 | 1e-6 | 5e-6 | 1e-6 | 5e-6 | 0.1 | 0.1 |
| $\theta$ for inference | 0.1 | 0.1 | 0.1 | 0.9 | 0.1 | 0.1 | 0.9 | 0.9 |
| Training Cost (Hour) | 7.2 | 5.0 | 2.5 | 8.1 | 7.0 | 1.5 | 5.7 | 3.0 |

Table 9: Performance with different $\lambda$. Threshold $\theta$ is set to 0.1.

| $\lambda$ | Remove Ratio | Latency(s) | Speedup | IS↑ | FID↓ | sFID↓ | Precision↑ | Recall↑ |
|---|---|---|---|---|---|---|---|---|
| 0 | 0/560 | 2.87 | 1.00× | 223.49 | 3.48 | 4.89 | 78.76 | 57.07 |
| 5e-7 | 129/560 | 2.55 | 1.13 × | 222.15 | 3.49 | 4.79 | 78.47 | 57.36 |
| 1e-6 | 176/560 | 2.45 | 1.17 × | 226.13 | 3.47 | 4.58 | 79.19 | 56.47 |
| 5e-6 | 248/560 | 2.28 | 1.26 × | 226.95 | 3.45 | 4.64 | 79.20 | 55.82 |
| 1e-5 | 300/560 | 2.15 | 1.33 × | 223.41 | 3.70 | 4.91 | 78.88 | 56.36 |
| 5e-5 | 404/560 | 1.92 | 1.49 × | 200.60 | 5.43 | 6.55 | 75.06 | 57.54 |
| 1e-4 | 460/560 | 1.79 | 1.60 × | 193.75 | 6.51 | 7.71 | 73.55 | 56.55 |

one that would affect is the $\lambda$ for training and the threshold $\theta$ for inference. We list in Table 8 the $\lambda$ we use that could reproduce the results in Table 1. Here the difference between DiT and U-ViT for $\lambda$ comes from the difference in implementation.

The results of using different $\lambda$ values are presented in Table 9. Note that $\lambda$ serves as the regularization strength to control the sparsity of the router, and thus there would not exist an optimal $\lambda$ for all settings. It functions as a trade-off between latency and quality, balancing the speed of inference with the fidelity of the generated images.

## C  Social Impact

The acceleration of diffusion transformers provides several positive social impacts, such as reducing the latency and resources required for deploying diffusion models. This enhancement improves the real-time applicability of diffusion transformers and promotes environmental sustainability. By making diffusion models more efficient, our method reduces the computational power needed for both training and inference, leading to lower energy consumption and a reduced carbon footprint. However, it is important to note that our method does not address privacy concerns, nor does it mitigate issues related to bias and fairness in diffusion models. These challenges remain when applying our method.

