# OpenReview forum: "Learning-to-Cache: Accelerating Diffusion Transformer via Layer Caching"
_NeurIPS.cc/2024/Conference — NeurIPS 2024 poster_

### Official Review · Reviewer_NP2z · 2024-06-14

**Soundness:** 3
**Presentation:** 3
**Contribution:** 2
**Rating:** 6
**Confidence:** 4

**Summary:**

This paper proposes a method to accelerate DiT model inference using layer caching strategy. By utilizing feature interpolation, the non-differentiable layer selection problem is transformed into a differentiable optimization problem. The routing matrix $\beta$ is learned to indicate whether the features of a certain layer at the current timestep can be reused from the cached features of the same position in the previous timestep. Extensive experimental results demonstrate the effectiveness of this method in accelerating DiT model inference and also shed light on the redundancy of layers in current DiT models.

**Strengths:**

1. Learning-to-Cache (L2C) transforms the non-differentiable layer selection problem into a differentiable optimization problem through interpolation, which is a clever transformation and forms the basis for optimizing the subsequent routing matrix $\beta$.
2. It is meaningful to explore how to apply feature caching mechanisms to DiT models for inference acceleration. The experimental results of the paper also demonstrate the effectiveness of the method. Compared with simply reducing NFE and previous feature caching methods, L2C achieves better performance.
3. This paper is well-organized and well-written;

**Weaknesses:**

My main concern lies in the scalability of this method:
1. L2C requires training for different DiT models and diffusion schedulers, which limits the potential applications of this method;
2. This paper reports experimental results on DiT and U-ViT series models, but does not experiment on text-to-image models based on the DiT architecture (e.g., Pixart-$\alpha$[1]). Is it because training on large-scale text-image pairs dataset is too costly?
3. The paper does not report specific training costs time;
4. Compared to the original inference process, will L2C increase additional memory due to routing matrix $\beta$ overhead and feature caching?

[1] Chen, Junsong, et al. "PixArt-$\alpha $: Fast Training of Diffusion Transformer for Photorealistic Text-to-Image Synthesis." arXiv preprint arXiv:2310.00426 (2023).

**Questions:**

see weakness

**Limitations:**

Limitations are discussed briefly, but could be touched upon in more detail.

---

> ### Author Rebuttal · Authors · 2024-08-07
>
> We sincerely appreciate the valuable feedback and constructive suggestions. Thanks so much for taking time and effort to review our paper.
>
> > **W1: L2C requires training for different DiT models and diffusion schedulers, which limits the potential applications of this method**
>
> Thank you for your insightful question. The training cost of L2C is low because only the router matrix is updated, leaving the model parameters unchanged. For example, in PixArt-XL-2 with a 512 resolution, only 840 parameters need to be trained, taking just 3 hours on 8 A5000 GPUs. We consider this training overhead to be relatively small; using a distillation approach to compress a small model typically requires significantly more resources, such as 4 A100 GPU days for BK-SDM [1].
>
> [1] BK-SDM: A Lightweight, Fast, and Cheap Version of Stable Diffusion. ECCV24.
>
> _________________
>
> > **W2: No experiment on text-to-image models based on the DiT architecture (e.g., Pixart). Is it because training on large-scale text-image pairs datasets is too costly?**
>
> Thanks for the valuable suggestion. Here we show the experimental results on PixArt-XL-2-512x512.
>
> * **Training Cost**:
> The training cost for this experiment is approximately 3 hours on 8 A5000 GPUs, using around 200,000 images for training. We utilized the first 200,000 samples from the SAM-LLaVA-Captions10M dataset. The router matrix converges and achieves optimal performance with 200,000 samples. We also tested with 400,000 samples but observed no performance gain, as the router does not change with additional training samples.
>
> * **Generation Quality**:
> We test our method on the validation set of COCO2014 (30k) and COCO2017 (5k). The results, shown in the table below, indicate that our method outperforms the approach using fewer steps in DPM-Solver. We provide some qualitative examples in Figure 5 of the attached PDF.
>
> * **What the router learns**:
> We have observed some intriguing patterns in the router, as illustrated in Figure 1 of the attached PDF. The cross-attention block displays significantly more temporal redundancy compared to other types of modules, such as self-attention and MLP. In addition, this router has the unique feature of not being cacheable at intermediate steps. We believe these specific patterns can also guide the future design of model architecture, helping to eliminate unnecessary computations.
>
> | Method     | NFE | Activate Layer Ratio in Cache Steps | Latency(s) | SpeedUp | Training Cost(h) | FID (COCO2017)↓ | FID (COCO2014)↓ |
> |------------|-----|----------------------|---------|---------|------------------|-----------------|-----------------|
> | DPM-Solver | 20  | 100\%                | 2.14    | 1.00x   | -                | 32.51           | 27.14           |
> | DPM-Solver | 14  | 100\%                | 1.51    | 1.41x   | -                | **33.79**           | **28.40**         |
> | L2C        | 20  | 31.3\%               | 1.52    | 1.41x   | 3.3              | **32.36**           | **27.39**           |
>
> _________________
>
> > **W3: The paper does not report specific training costs time;**
>
> We apologize for missing this important experimental detail. Here, we report the training cost for each experiment. We use ImageNet for training, and in all cases, convergence is achieved in less than one epoch. The experiments are conducted on 8 A5000 GPUs. We would add this training cost to our next version of the manuscript. Thanks so much for pointing this out.
>
> | Model                | DiT-XL/2 | DiT-XL/2 | DiT-XL/2 | DiT-XL/2 | DiT-L/2 | DiT-L/2 | U-ViT-H/2    | U-ViT-H/2    |
> |----------------------|----------|----------|----------|----------|---------|---------|--------------|--------------|
> | NFE                  | 50       | 20       | 10       | 50       | 50      | 20      | 50           | 20           |
> | Resolution           | 256      | 256      | 256      | 512      | 256     | 256     | 256          | 256          |
> | Sampler              | DDIM     | DDIM     | DDIM     | DDIM     | DDIM    | DDIM    | DPM-Solver-2 | DPM-Solver-2 |
> | Training Cost (Hour) | 7.2      | 5.0      | 2.5      | 8.1      | 7.0     | 1.5     | 5.7          | 3.0          |
>
> _________________
>
> >  **W4: Compared to the original inference process, will L2C increase additional memory due to routing matrix overhead and feature caching?**
>
> * For Routing Matrix:
>
> The routing matrix has a very small number of parameters, calculated as $steps \times layers \times block\\_per\\_layer$. In our experiments, the largest router, used for DiT-XL/2 with 50 sampling steps, contains 1,400 parameters ($25 \times 28 \times 2$), resulting in an extra memory overhead of just 2.8KB (using FP16 for inference).
>
> * For Feature Caching:
>
> Yes, L2C needs extra overhead for feature caching. As the cache in the computer system and KV-Cache in LLM, L2C trades space for time by storing intermediate results in VRAM, leading to additional memory overhead. Below is the additional overhead observed in the DiT model. We believe there is still room for optimization of the memory overhead in feature caching. Thanks for raising this critical issue.
>
> | Method | Memory |
> | -- | -- |
> | DiT-XL/2 | 3905 MiB |
> | DiT-XL/2 with L2C| 4831MiB |

---

> > ### Author Response · Authors · 2024-08-12
> >
> > Dear Reviewer NP2z,
> >
> > We sincerely appreciate the time and effort you have dedicated to reviewing our work. We greatly appreciate your thoughtful review of our work and we are looking forward to hearing your feedback.
> >
> > To address your concerns regarding the scalability of our method, we have conducted additional experiments, including:
> >
> > 1. Experimental results on PixArt, alongside the comparison results with the few-step DPM-Solver.
> > 2. Showing that the training cost for the router remains within an acceptable range.
> >
> > We are grateful for your attention to our rebuttal and are committed to addressing any additional concerns you might have.
> > Thank you once again for your thoughtful consideration.
> >
> > Best regards,
> > Authors of submission 1630

---

> > > ### Comment · Reviewer_NP2z · 2024-08-12
> > > **Official Comment by Reviewer NP2z**
> > >
> > > I appreciate the authors' rebuttal and extra experiments, most of my concerns have been resolved. I will keep my positive score.

---

> > > > ### Author Response · Authors · 2024-08-12
> > > >
> > > > Thank you so much for your valuable update. We will follow your comments to revise our submission.

---

> > > > > ### Comment · Reviewer_NP2z · 2024-08-13
> > > > >
> > > > > I have raised my score and I wish you good luck :)

---

> > > > > > ### Author Response · Authors · 2024-08-13
> > > > > >
> > > > > > Thanks so much for the quite encouraging feedback. We will continue to polish our draft, following your advice. Thank you again for your time and effort in reviewing our submission

---

### Official Review · Reviewer_faWt · 2024-07-09

**Soundness:** 3
**Presentation:** 3
**Contribution:** 3
**Rating:** 6
**Confidence:** 2

**Summary:**

This paper introduces L2C, a novel approach that dynamically caches computations in diffusion transformers, significantly reducing the computational load. L2C leverages the repetitive structure of transformer layers and the sequential nature of diffusion, optimizing caching decisions to produce a static computation graph. Experimental results show that L2C outperforms existing methods like DDIM and DPM-Solver, as well as prior cache-based techniques, at the same inference speed.

**Strengths:**

- The writing is easy-to-follow


- The motivation is strong. There exists many redundancy and the authors bypass it in a smart way

**Weaknesses:**

I am not an expert in this area. There is no obvious weakness as far as I can tell

**Questions:**

See Weaknesses

**Limitations:**

See Weaknesses

---

> ### Author Rebuttal · Authors · 2024-08-07
>
> Thank you very much for your review of our manuscript. We appreciate your time and effort in evaluating our work.
>
> It is encouraging to hear that you like our work and that no obvious weaknesses have been found. If you have any questions where you think further detail or explanation might be beneficial, we would be happy to address them.
>
> Thank you again for your valuable time and feedback.

---

### Official Review · Reviewer_FT6Q · 2024-07-11

**Soundness:** 4
**Presentation:** 4
**Contribution:** 3
**Rating:** 6
**Confidence:** 5

**Summary:**

The paper presents Learning-to-Cache (L2C), a method to accelerate diffusion transformers' inference by caching redundant computations across timesteps. A learnable router dynamically determines which layers can reuse calculations from previous timesteps. L2C can eliminate up to 93.68% of computations in specific steps (46.84% overall) in models like U-ViT-H/2 with minimal performance loss.

**Strengths:**

- The proposed caching method for the diffusion process is novel and provides acceptable speed-up without requiring retraining or fine-tuning of the model weights, only adding a few learnable parameters.
- The paper introduces a straightforward learnable optimization process to identify redundancies in the diffusion process.
- The proposed router is time-dependent but input-invariant, enabling the formation of a static computation graph for inference.
- The paper is well-written and easy to understand. The figures and result tables are easy to follow and comprehensive, offering clear visual representations and supporting the text.

**Weaknesses:**

- The paper's contribution is incremental, primarily introducing a learnable router that determines what computations to reuse from previous timesteps. It would benefit from offering more innovations and deeper insights to significantly advance the field.
- The improvement in speed-up is very similar to DeepCache, which does not require any training.
There are some minor mistakes in the text, such as a typo in line 233 where "maximum" is misspelled as "mamimum."

**Questions:**

- Step distillation techniques, which typically involve few or just one step, have successfully accelerated the diffusion process with minimal impact on output quality. Can the proposed model be integrated with distilled models to achieve additional speed-up on top of the improvements gained from distillation?
- You have proposed training only the router parameters while freezing the model parameters. Is there any benefit to fine-tuning and optimizing both the model parameters and the router parameters together?
- What are the implications if the diffusion main loss function is used to train the router?

**Limitations:**

Since the goal is to speed up inference, it should be discussed whether this method can be combined with other acceleration techniques, such as step distillation.

---

> ### Author Rebuttal · Authors · 2024-08-07
>
> We extend our gratitude for your insightful feedback and suggestions
>
> > **W1: The paper's contribution is incremental. It would benefit from offering more innovations and deeper insights**
>
> We greatly value your suggestion that we need to offer more profound insights in this paper. Beyond introducing a new method, we aim to share the following key insights to advance the field:
>
> (1) **Theoretical explanation of the caching-based method**: In Section 3.2\&3.3 and Appendix, we reveal that fast samplers and cache mechanisms represent different levels of reuse. Unlike previous heuristic-based approaches like DeepCache, our work establishes a theoretical connection and build the clear relationship between cache and fast solvers. This not only clarifies their relationship but also enhances the reliability of caching-based methods by providing theoretical support.
>
> (2) **Insights from the special patterns in the learned routers**: L2C reveals intriguing patterns in the learned routers (refer to Figures 1, 2, 3 and 4 in the attached PDF).  In U-ViT (Figure 2), the intermediate layers can be entirely cached, whereas the early and later layers play a crucial role. We also add one experiment on PixArt-XL-2 and for PixArt, the temporal redundancy in the cross-attention layer (Figure 1.b) is notably pronounced, with over 94\% of layers (265 out of 280) being eligible for caching.These findings can be leveraged not only to accelerate model inference but also to guide the design of model architectures.
>
> To summarize, we aim to move beyond heuristic methods in designing the cache mechanism, incorporating performance guarantees and providing more theoretical interpretability for this special cache mechanism in accelerating the diffusion models. This has not been explored before and we think is important for this area.
> ___
>
> > **W2: The improvement in speed-up is very similar to DeepCache.**
>
> Thanks for your insightful quesion. We compare our approach with DeepCache, focusing primarily on **generation quality** under the same acceleration ratio. Our method improves the FID from 2.70 to 2.64, using the same model and parameters, indicating a superior caching mechanism compared to DeepCache. Additionally, DeepCache is limited to U-ViT due to it is bound to the u-shaped structure, making it inapplicable to models like DiT and PixArt. However, L2C is versatile and can be applied to all these models.
>
> | Methods | NFE | Latency(s) |Speedup |FID↓ |
> |--|--|--|--|--|
> | DPM-Solver | 20 |7.60 |1.00x | 2.57 |
> | DeepCache | 20 |4.68 |1.64x | 2.70 |
> | Faster Diffusion | 20 |5.95 | 1.29x | 2.82 |
> | L2C (ours) | 20 | 4.62 |1.67x | 2.64 |
> ___
>
> > **Q1\&Limitation: Can the proposed method be combined with step distillation?**
>
> Thank you for your constructive suggestion. We built our algorithm on the distilled PixArt-XL-2-512x512, employing a 4-step LCM scheduler. The table below presents our results, which are compared against the 3-step LCM. The learned router is visualized in Figure 4 of the attached PDF. From our experiments, our method achieves approximately 1.28x acceleration, successfully caching 78 out of 168 blocks. Consistent with the results on PixArt with 20 NFEs, cross-attention remains the most redundant component in the denosing sequence.
>
> | Method | NFE | Cache Layer Ratio | Latency(s) | Speedup | FID (COCO2017)↓ | FID(COCO2014)↓ |
> |--|--|--|--|--|--|--|
> | PixArt-XL-2 + LCM | 4 | - |0.96 | 1.00x | 34.52 | 29.60|
> | PixArt-XL-2 + LCM | 3 | - |0.73 | 1.32x | **34.77**| **29.72**|
> | PixArt-XL-2 + LCM + L2C|4|78/168 |0.75|1.28x | **34.45** | **29.26** |
> ____
>
> > **Q2: Optimizing both the model parameters and the router parameters**
>
> Thanks for your great suggestion. We show the comparison results if optimizing the model parameters and the routers together:
>
> | Method  | Cache Layer Ratio | Latency(s) | Speedup | IS  | FID↓  | sFID↓  | Precision | Recall | Training Cost↓ |
> |--|--|--|--|--|--|--|--|--|--|
> | DDIM20 | - | 2.87 |  1.00x | 223.49 | 3.48   | 4.89  | 78.76 | 57.07  | - |
> | DDIM15 | -  | 2.17| 1.32x | 205.97 | 5.07   | 6.07  | 76.26 | 55.65  |
> | Router | 333/560 | 2.09 | 1.37x | 219.85 | **4.01**   | 5.22  | 78.28 | 55.73  | ~40 GPU Hour |
> | Router&Model   | 331/560 | 2.09 | 1.37x   | 213.34 | **3.87**  | 5.00  | 77.83  | 57.84 | ~66 GPU Hour |
> |||
> | DDIM10 | - | 1.43 | 1.00x | 158.31 | 12.38  | 11.22 | 66.78  | 52.82  | - |
> | DDIM9  | - | 1.29 | 1.11x | 140.85 | 16.57  | 14.21 | 62.28   | 49.98 | -   |
> | Router  | 107/280 | 1.17 | 1.22x| 147.77 | **14.63**  | 10.96 | 64.30  | 51.65  | ~20 GPU Hour  |
> | Router&Model   | 107/280 | 1.17 | 1.22x| 134.87 | **14.02** | 12.43  | 64.92  | 52.37 | ~41 GPU Hour |
> ||
>
> The conclusion is that, yes, optimizing both the model and the router enhances the model's performance, as evidenced by slight improvements in FID and sFID. However, this optimization process requires additional time to train and fine-tune both components. We will include this experiment in our revised manuscript and thanks again for your inspiring suggestion.
> ____
>
> > **Q3: what if the diffusion main loss function is used to train the router?**
>
> Thank you for your insightful question. We have adjusted the training loss accordingly, and the results are presented in the table below. We selected baselines with a very similar acceleration ratio for comparison. The results indicate that using the original diffusion loss makes it more challenging for the router to learn, resulting in slightly worse image quality compared to the distillation loss used in our submission.
>
> | Method | Cache Layer Ratio   | Latency | Speedup | IS | FID↓| sFID↓|Precision | Recall |
> |--|--|--|--|--|--|--|--|--|
> | DDIM-20 | -  | 2.87| 1.00x   | 223.49 | 3.48 | 4.89   | 78.76 | 57.07  |
> | DDIM-15  | -  | 2.17 | 1.32x   | 205.97 | 5.07 | 6.07   | 76.26| 55.65  |
> | Original Loss | 295/560 | 2.18 | 1.32x   | 217.23 | **3.97** | 5.11 | 77.83  | 57.05  |
> | Our Loss | 300/560 | 2.16 | 1.33x   | 223.41 | **3.70** | 4.91 | 78.88  | 56.36  |

---

> > ### Author Response · Authors · 2024-08-12
> >
> > Dear Reviewer FT6Q:
> >
> > Thank you for your valuable feedback on our work. Your constructive comments on our work are invaluable, and we genuinely hope to get feedback from you.
> >
> > Regarding the weaknesses you mentioned, we include the corresponding experiments as follows:
> >
> > 1. Experimental results when **applying Learning-to-Cache to 4-step LCM on PixArt**.
> > 2. Generation quality compared with other cache-based methods (DeepCache, Faster Diffusion)
> > 3. Experimental results when optimizing both the model parameters and the router parameters
> > 4. Experimental results using the original diffusion loss.
> >
> > Your feedback is incredibly important to us, and we sincerely thank you for considering our rebuttal. We are more than happy to discuss them if you have any further concerns or questions.
> >
> > Thank you again for your time and effort to review our work and looking forward to your response.
> >
> > Best Regards,
> >  Authors of submission 1630

---

> > > ### Comment · Reviewer_FT6Q · 2024-08-12
> > > **Increasing the Score**
> > >
> > > I want to thank the authors for their rebuttal and answering my questions. I will raise my score to WA.

---

> > > > ### Author Response · Authors · 2024-08-13
> > > >
> > > > Thank you for reviewing our paper and providing valuable feedback. We're glad our rebuttal addressed your concerns, and we'll include the suggested experiments in the revised manuscript. Thanks again for your time and effort.

---

### Author Rebuttal · Authors · 2024-08-07

Dear Chairs and Reviewers,

We deeply appreciate your thoughtful comments and the time you have dedicated to reviewing our paper. Attached is a pdf containing the following:

* Visualizations of learned routers in different models
* Generated images compared with the baseline.

We look forward to the opportunity to discuss this further with you. Many thanks for your kind attention.

Best regards,
Authors of submission 1630

---

### Comment · Area_Chair_6AMM · 2024-08-10
**Rebuttal Discussion**

Dear Reviewers,

This is your AC. The authors have provided a response to the comments. Please respond to the rebuttal actively.

Best,
AC

---

### Decision · Program_Chairs · 2024-09-25

**Decision:**

Accept (poster)

**Comment:**

The paper presents Learning-to-Cache (L2C), a method to accelerate diffusion transformers' inference by caching redundant computations across timesteps. Particularly, by leveraging the repetitive structure of transformer layers and the sequential nature of diffusion, L2C dynamically caches the computations in diffusion transformers for saving computational load. The main concerns were well addressed during rebuttal. I recommend to accept the manuscript and the authors should incorporate the discussions in the final version.